# Magnetoelastic Sensor Optimization for Improving Mass Monitoring

**DOI:** 10.3390/s22030827

**Published:** 2022-01-22

**Authors:** William S. Skinner, Sunny Zhang, Robert E. Guldberg, Keat Ghee Ong

**Affiliations:** Knight Campus for Accelerating Scientific Impact, University of Oregon, Eugene, OR 97403, USA; wss@uoregon.edu (W.S.S.); sunnyz@uoregon.edu (S.Z.); guldberg@uoregon.edu (R.E.G.)

**Keywords:** magnetoelastic, magnetostrictive, sensor, monitoring, wireless, mass, geometry, resonance

## Abstract

Magnetoelastic sensors, typically made of magnetostrictive and magnetically-soft materials, can be fabricated from commercially available materials into a variety of shapes and sizes for their intended applications. Since these sensors are wirelessly interrogated via magnetic fields, they are good candidates for use in both research and industry, where detection of environmental parameters in closed and controlled systems is necessary. Common applications for these sensors include the investigation of physical, chemical, and biological parameters based on changes in mass loading at the sensor surface which affect the sensor’s behavior at resonance. To improve the performance of these sensors, optimization of sensor geometry, size, and detection conditions are critical to increasing their mass sensitivity and detectible range. This work focuses on investigating how the geometry of the sensor influences its resonance spectrum, including the sensor’s shape, size, and aspect ratio. In addition to these factors, heterogeneity in resonance magnitude was mapped for the sensor surface and the effect of the magnetic bias field strength on the resonance spectrum was investigated. Analysis of the results indicates that the shape of the sensor has a strong influence on the emergent resonant modes. Reducing the size of the sensor decreased the sensor’s magnitude of resonance. The aspect ratio of the sensor, along with the bias field strength, was also observed to affect the magnitude of the signal; over or under biasing and aspect ratio extremes were observed to decrease the magnitude of resonance, indicating that these parameters can be optimized for a given shape and size of magnetoelastic sensor.

## 1. Introduction

Magnetoelastic sensors are ideal for remote query of environmental conditions in closed and controlled systems because they can be fabricated and scaled into different shapes and sizes, functionalized with a wide array of chemical/biological sensing elements, and remotely activated and interrogated [1,2]. The operation of a magnetoelastic sensor is based on its magnetostrictive behavior, which allows it to undergo cyclic mechanical vibrations at its resonance frequency when exposed to an alternating magnetic field. The mechanical resonance in turn causes the sensor to generate a secondary magnetic flux that can be detected independently (Figure 1). When a mass is loaded onto the sensor, the resonance frequency, magnitude of resonance, and signal phase at the resonance change due to the damping to the sensor’s mechanical resonance. This change in resonance character and quality can be used to quantify the mass loaded onto the sensor’s surface. Magnetoelastic sensors have been widely investigated for remotely monitoring environmental parameters, including physical quantities such as humidity, pressure, and the viscosity and density of liquids [2,3,4]. Detection and monitoring of chemical analytes and biologics with magnetoelastic sensors typically require functionalizing the sensor with a coating that gains or loses mass as a result of the presence or activity of the chemical/reaction/biologic being detected. This coating can be an enzyme or molecule that specifically binds with an analyte, a coating that absorbs/desorbs water as pH changes, or molecules that are specifically consumed during reactions/metabolism. Functionalization of magnetoelastic sensors has increased their popularity as a platform for label-free detection [5] and has facilitated the detection and measurement of chemical analytes, such as pH [2], ammonia [2], carbon dioxide [2], calcium oxalate precipitate [6], and octachlorostyrene [7], and biologicals and biotoxins such as *E. coli*, Staphylococcus enterotoxin B, and other endotoxins [8,9,10,11,12,13].

Magnetic sensing techniques are not limited to the detection of biologics based on mechanical resonance due to mass loading. For instance, Quandt et al. reported a magnetoelectric sensor based on composite magnetoelastic and piezoelectric thin films with detection limits as low as 2.6 pT/Hz^1/2^ [14,15]. These results would suggest that these materials could be suitable for biomagnetic sensing applications, such as magnetocardiography, magnetoencephalography, and other magnetic immunoassays which have historically been limited to the most sensitive magnetic materials, such as superconducting quantum interface devices (SQUIDs) [16,17]. Similarly, García-Arribas et al. reported the fabrication of magneto-impedance, magneto-elastic, and magneto-electric sensors for detecting small magnetic fields by layering composites of soft magnetic alloys with other materials [4].

Magnetoelastic sensors are typically fabricated from amorphous ferromagnetic materials such as Fe_40_Ni_38_Mo_4_B_18_ (Metglas 2826 MB) or Fe_81_B_13.5_Si_3.5_C_2_ (Metglas 2605SC). These materials exhibit strong magnetostrictive behavior and have high magnetoelastic coupling coefficients so that energy conversion from elastic to magnetic energies, and vice versa, are highly efficient [18,19,20]. Sensors made from these materials can be excited by a time-varied magnetic field, which results in cyclic stretching/relaxation of the material. Due to its high magnetoelastic coupling, the resonating material generates a strong secondary magnetic flux, which can be remotely detected and tracked with an induction coil. Similar to all mechanical resonating bodies, the resonance of the magnetoelastic sensors is affected by its geometry, local environment, and mass-loading profile. Therefore, understanding and optimizing these parameters can lead to an improvement in the performance of these sensors by increasing their signal strength, detection range, and sensitivity towards mass loading.

Optimizing the magnetoelastic sensor’s sensitivity towards mass loading is particularly important in the context of biological/chemical sensing, which often requires the detection of very small changes in mass in the nanogram range or even lower. Previous work by a number of investigators has demonstrated the biological sensing capability of this technology with L929 fibroblast cells [21], breast cancer cells [22], and bacteria [9,11,12,13,23,24,25,26]. However, further improvement of the magnetoelastic sensor technology will be required to realize a practical mass-based biological/chemical sensing system. In particular, understanding how variations in the sensors’ geometry, position, and orientation relative to the detection system can affect the measurement quality of the resonance spectrum. Understanding and optimizing these parameters will be critical for developing this technology for use in dynamic and volumetric environments, such as a bioreactor. Multiple investigators have conducted similar studies in the pursuit of optimizing these sensors for mass detection: Sagasti et al. recently published a study investigating the effects of size and aspect ratio on several parameters related to the resonance and magnetic activity of the sensor [27] as well as another study determining the effects of thin polymeric coatings on the resonance behavior of the sensor [28]; Saiz et al. recently published multiple experiments detailing the effects of geometry on sensor sensitivity, including triangular and rhombohedral shapes [29,30]; Pacella et al. have also investigated triangular-shaped sensors and non-uniform coatings as a means of increasing the sensitivity of magnetoelastic sensors [1]. This study was focused on investigating parameters relevant to the employment of the sensors in volumetric and dynamic systems. We investigated multiple approaches for optimizing the sensors’ performance by altering their shapes, sizes, and orientations with respect to the applied magnetic fields. Additionally, variations in sensitivity across the sensors’ surface were investigated, and sensors of different geometries were rotated in the detection coil to understand how two-dimensional angular orientation impacted the sensors’ resonance behavior. We also investigated the dimensional aspect ratio’s impact on sensor response and determined the optimal magnetic bias field for sensors of different sizes and aspect ratios.

## 2. Materials and Methods

### 2.1. Sensor Fabrication

Sensors were mechanically sheared from commercially available magnetoelastic material, Metglas 2826 MB (Metglas Inc., Conway, SC, USA) [31]. The material came in a rolled, thick-film strip that was 12.7 mm wide and 29 µm thick (saturation magnetization = 0.39 T, coercive force <50 A/m). After sensors were sheared to shape, they were annealed at 125 °C for 2 h. The annealing step is critical, as mechanically shearing the material can introduce localities of stress and non-uniformity of features, particularly at the edge of the sensor, where the sensor is most prone and sensitive to defects (Figure 2). Annealed sensors were then coated with a 10 µm Parylene-C conformal layer via a commercially available vapor deposition coating system (PDS 2010 Labcoater, SCS, Indianapolis, IN, USA). The coated sensors were then treated with oxygen plasma in a reactive etching system (March Jupiter II RIE system, Nordson March, Concord, CA, USA) for 30 s at 100 Watts. This technique was previously developed by Holmes et al. and employed in the fabrication of sensors for the detection of L929 fibroblast cells [21,32].

### 2.2. Detection System

As illustrated in Figure 3 and Figure 4, the magnetoelastic sensors were interrogated with a system consisting of two concentric custom-wound solenoids. The outer solenoid was connected to a DC power source to supply a biasing current between 1 A and 2.5 A, resulting in the generation of a DC magnetic field between 1.4 × 10^3^ A/m and 3.6 × 10^3^ A/m. The inner coil, wound with 36-gauge magnet wire, was connected to a network analyzer (Keysight ENA Network Analyzer E50618, Keysight Technologies, Santa Rosa, CA, USA), which recorded the resonance data from the sensors by measuring the S11 parameter at the coil terminal. In essence, the network analyzer generated a time-varied, broadband voltage signal and sent it to the detection coil. Following this, the reflected signal from the coil was measured and compared against the incident signal to determine the coil’s impedance, and thus, the magnitude and phase of signals recorded by the coil. The resonance of the sensor can be measured by this technique because the magnetic flux generated by the sensor inside of the detection coil modulated the coil’s impedance via inductive coupling (Figure 3).

The detection system and cable were calibrated manually with a digital open-short-load calibration kit (ECal module N7550A, Keysight Technologies, Santa Rosa, CA, USA). After the cable was calibrated, the coils were connected, and background measurements were recorded prior to inserting the sensors. Next, a single sensor was placed in the rear well of a 2-well chamber slide (Nunc Lab-Tek II Chamber Slide, Thermo Fisher Scientific, Waltham, MA, USA) on top of a 3D printed stage designed to orient the sensor without restraining its mechanical resonance. The chamber slide was placed inside the detection solenoid connected to the network analyzer to record the resonance spectrum of the sensor inside. Due to the length of time between measurements, background measurements were collected for each instance of measurement to establish an accurate baseline signal.

### 2.3. Determining Mass-Loading Sensitivity at Various Locations of the Sensor

Three rectangular sensors (12.7 mm × 5 mm) were cleaned with acetone and weighed with a microbalance. Following cleaning, the resonance spectrum was collected for each sensor. A mass load was applied to the sensors via deposition of a drop of UV-curable adhesive (Dymax MD^®^ Medical Adhesive 1072-M, Dymax, Torrington, CT, USA). The interface between the adhesive drop and the sensor’s surface had a diameter of 1 mm or less. As illustrated in Figure 5, eight locations, selected based on symmetry, were designated for the load placement.

Three approximate drop sizes were utilized: a “small” drop with a diameter of 0.6 mm ± 0.1 mm, a “medium drop” with a diameter of 0.8 mm ± 0.1 mm, and a “large” drop with a diameter of approximately 1 mm ± 0.1 mm. Drops were deposited and cured under UV light for a period of 300 s. The sensors were then weighed, followed by measuring their resonances. These sensors were then thoroughly cleaned with acetone prior to the next deposition. The frequency shift per unit mass applied was calculated for each load placement and averaged among three trials.

### 2.4. Effects of Rotation on the Resonance Spectrum for Sensors of Different Shapes

Sensors of different shapes were fabricated from Metglas 2826 MB (a roll of long Metglas ribbon with a width of 12.7 mm) to analyze the effect of shape on the sensors’ performance at different orientations from the applied magnetic fields. Rectangular (aspect ratio of 2.5), square, and equilateral triangle sensors were cut to 12.7 mm along the longest dimension in their normal orientation (length of rectangle and square, height of the equilateral triangle). This normal orientation was defined as the longest dimension of the shape, which was also coincided with the longitudinal dimension of the Metglas ribbon and the applied magnetic fields. The Metglas material exhibits homogenous striations parallel with its longitudinal dimension. These striations are a result of the material’s manufacturing process and can have a small influence on the resonance behavior of the sensor. Typically, having the striations aligned with the applied magnetic fields resulted in slightly sharper and stronger resonance peaks. These sensors were measured in their normal orientation (0°, Figure 6) and other orientations by rotating the sensors within the coil as illustrated in Figure 6. The maximum angular displacement investigated for the rectangular, square, and triangular sensors were 90°, 180°, and 120°, respectively.

### 2.5. Determination of Optimal DC Bias Field Magnitude for Rectangular Sensor Resonance

Three sets of five sensors were fabricated into sizes of 6 mm × 2.4 mm, 9 mm × 3.6 mm, and 12.7 mm × 5 mm, such that all sensors had an aspect ratio of 2.5 (L/W). These sensors were subjected to a range of magnetic DC bias fields from 1.4 × 10^3^ A/m to 3.6 × 10^3^ A/m generated from a power supply with DC currents ranging from 1 A to 2.5 A. The optimal bias field magnitude for each set of sensors was defined as the field magnitude which gave the sharpest resonance peak with the greatest resonance magnitude for each sensor. To determine the optimal bias field, the DC supply current was varied in 0.1 A increments, beginning at 1 A. At each current increase, the resonance spectrum of the sensor was analyzed via the network analyzer until the increases in field magnitude began to result in resonance peaks with decreasing magnitude. Beyond the optimal value, an over-biased sensor behaved similarly to an under-biased sensor, in that the resonance peak decreased in quality and magnitude. The under-biased and over-biased resonance behavior of the sensors were used to identify the optimal DC bias field value which resulted in a maximized resonance magnitude from the sensors.

### 2.6. Effect of Aspect Ratio on Rectangular Sensor Resonance

The goal of this investigation was to identify the optimum aspect ratio for maximizing the magnitude of the sensors’ response. Two sets of sensors, of length 9 mm and 12.7 mm, were fabricated into rectangles with four different aspect ratios (1, 1.5, 2.5, and 4) by changing their widths to 12.7 mm, 8.5 mm, 5 mm, and 3.2 mm, respectively. The sensors were analyzed with bias field values optimized for resonance peak magnitude and sharpness. This was accomplished using the formerly described method of analyzing a range of bias currents and selecting an optimal current based on under- and over-biased resonance behavior. The results were normalized against the surface area of the samples.

## 3. Results and Discussion

### 3.1. Mass-Loading Sensitivity at Various Locations of the Sensor

The rectangular sensors (*n* = 3) used in this experiment were expected to exhibit lengthwise mechanical resonance (longitudinal mode). Free-standing sensors were expected to have a node (minimal mechanical vibration) in the center of the material, with the most sensitive regions expected at the distal extremes. Analysis of the collected data indicated that the rectangular sensors fit the expected model: there was a region of low sensitivity at the center of the sensor (Figure 7A,B). If no point on the sensor is fixed and the sensor is symmetrical, the greatest resonance displacement happens at the edges furthest from the center. Hence, when the magnetoelastic sensors resonated, the maximum displacement occurred at the edges furthest from the center. As indicated in Figure 7, these regions exhibited the strongest resonance signals, which were responsible for the increased sensitivity relative to other regions.

### 3.2. Effects of Rotation on the Resonance Spectrum of Different Shapes

The rectangular sensors showed the greatest variation in sensor response per degree of rotation of all geometries explored (Figure 8A). This was anticipated, as the rectangular sensors resonated with a longitudinal mode along the longest portion of the sensor; for the rectangle, this mode exhibits maxima at 0° and 180°. This corresponds with the expected behavior, as a rectangle’s longest side can only be aligned with the direction of the excitation field at two orientations. However, another phenomenon related to the geometry of the rectangular sensors was also observed. Under a strong DC bias field, the rectangular sensors physically rotated and aligned to the direction of the applied DC bias field, conveniently positioning themselves in the optimum orientation for detection. It could be useful to take advantage of this phenomenon in dynamic and volumetric environments, where the sensors could self-orient to their optimal position for measurement.

The equilateral triangle-shaped sensors also resonated along the direction of the applied magnetic fields (longitudinal mode). Each triangle has 6 orientations in which the longest portion of the shape is aligned with the excitation field: 0°, 60°, 120°, 180°, 240°, and 300°. As a result of the increased number of orientations with aligned modes (in other words, the triangle’s higher degree of rotational symmetry), the equilateral triangle-shaped sensors exhibited less variance in response per degree of rotation than the rectangular sensors (Figure 8B).

The square-shaped sensors exhibited the least variance in response per degree of rotation (Figure 8C). The square-shaped sensors also appeared to resonate with multiple modes (Figure 9). Specifically, the response signal at 0° rotation showed 3 resonance peaks for the square sensors. However, at other orientations, one or two of the peaks disappeared and/or appeared dampened while the third resonance peak remained mostly unchanged. Analysis of the resonance frequency and response behavior at different orientations led to the identification of one of the resonance peaks that disappeared at certain orientations as the longitudinal mode (similar to the rectangular sensors). The peak that persisted in all orientations (about 210 kHz) is the result of one or multiple modes of resonance that are similar to the radial resonance of a circular membrane/diaphragm [33], as the peak shows relatively minimal variation, even at orientations that had significantly dampened the other two resonance peaks (Figure 9).

Analysis of this study indicates that sensor geometry can be used to reduce the change in sensor response due to a rotational change relative to the direction of the magnetic interrogation field. These results also suggest that optimization of sensors that resonate with a longitudinal mode and radial-like mode might be pursued separately, as the advantages and disadvantages of these modes are mostly exclusive to one another. Sensors with longitudinal resonances can potentially be oriented with the bias field, but they exhibit the greatest change in response to misalignment from the optimal orientation. Sensors with radial-like resonance could exhibit a consistent resonance response at any orientation confined to a particular plane; however, this mode of resonance may not exhibit the same level of performance as the longitudinal mode sensors of similar dimensions. In future research, sensors might take on a variety of geometries: circular sensors, star-like sensors that are round with several triangular points along the edge, and even octagonal sensors could be promising candidates for sensors that exhibit resonance modes similar to the vibration of a circular membrane/diaphragm.

### 3.3. Optimization of the DC Bias Field

The DC bias field is critical for increasing the amplitude of the sensors’ signals to a suitable level for detection. To boost the sensor’s signal, a bias field (in the form of a DC magnetic field) is required, which maximizes the sensor’s magnetostrictive behavior by partially aligning the magnetic moments within the sensor. When sensors are under-biased, there are significant cancellation effects in the rotation of magnetic moments during resonance, resulting in low net sensor vibration. When sensors are over-biased, the oversaturated magnetic moments are too strongly aligned to the bias field, resulting in minimal rotation under the agitation of the AC magnetic field. The results of the bias field optimization experiment indicated that an optimal DC magnetic bias field exists for a given sensor shape and is different for sensors of different dimensions. The results also indicated that it may be possible to predict the optimal bias field for a sensor of a particular geometry and aspect ratio if the optimal value is known for other dimensions of the same shape (Figure 10). In general, smaller sensors required a bias field of greater magnitude to achieve maximum magnetic flux output.

### 3.4. Effect of Aspect Ratio on the Sensor’s Resonance

Of the aspect ratios investigated, the ratio of 1.5 resulted in a maximized magnitude of response for the rectangular sensor shapes of both lengths (Figure 11A,B). However, after the data were normalized for differences in surface area between the two sets, as the aspect ratio increased, so too did the signal strength per unit area, with diminishing returns (Figure 12A,B). Narrower sensors exhibited more efficient field generation, as the net magnetization of these is stronger for a given excitation field due to their smaller demagnetization factor [34]. While aspect ratios around 1.5 exhibited the most optimum balance between the sensors’ self-dampening and field output, normalizing the results against the surface area for each aspect ratio revealed that an aspect ratio of higher than 1.5 emitted more magnetic flux per unit area than an aspect ratio of 1.5. While the 1.5 aspect ratio sensors host a greater surface area for load attachment, higher aspect ratio sensors exhibited more efficient field generation per unit area (Figure 12A,B). Consideration of these results suggests that aspect ratios of 1.5 could be optimal for general applications, but higher aspect ratios may be more desirable if miniaturized sensors are needed. We should also consider investigating other shapes, such as the rhombus, that can take advantage of increasing edge/corner surface area while still maintaining a predictable longitudinal resonance behavior. Saiz and coworkers recently published information regarding the sensitivity of the rhombus geometry related to other shapes. The results indicated that the rhombus exhibited more regions of higher sensitivity than rectangles or triangles of similar dimension/length [30]. In addition to considering other shapes and a more precise investigation, the relative sensitivity of all of these shapes should be taken into consideration in the determination of an optimized aspect ratio for rectangular sensors.

## 4. Conclusions

Optimization of the geometry of the sensors and measurement conditions will be key factors in the development of a biological/chemical detection system for use in dynamic and volumetric environments. In the case of mass-based sensing, the ideal shape for the sensors should be a geometry that takes advantage of the knowledge surrounding sensitivity and response magnitude. Sensors with resonance modes similar to the vibration of a circular membrane/diaphragm exhibited the least variance in resonance frequency when the direction of magnetic fields change. For rectangular sensors, there was a specific aspect ratio which resulted in an optimized response from the sensors, with each shape and size requiring a different magnitude of DC bias field to achieve an optimized resonance spectrum.

## Figures and Tables

**Figure 1 sensors-22-00827-f001:**
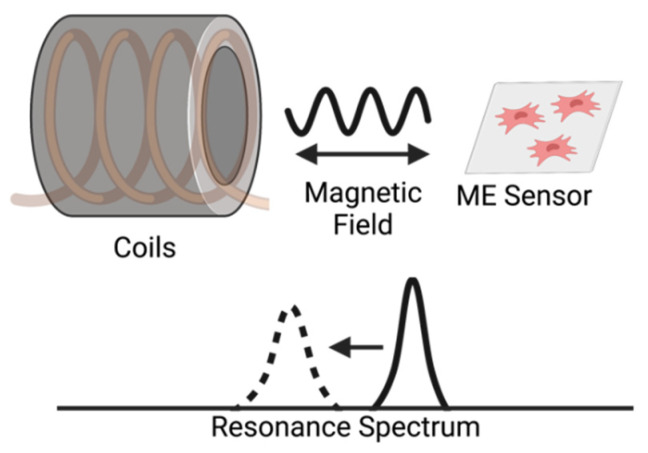
Illustration of the detection of a magnetoelastic sensor by measuring the change in its resonance spectrum. The magnetoelastic sensor and detection coil are inductively coupled, which records the sensor’s resonance spectrum by measuring the coil’s impedance spectrum.

**Figure 2 sensors-22-00827-f002:**
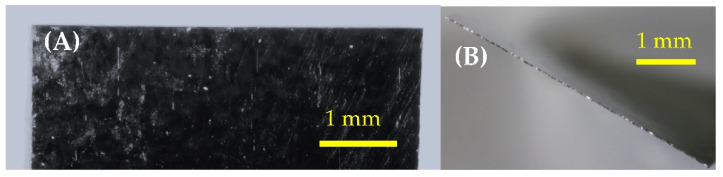
Microscopic images of the edge/corner profiles (**A**) and edge (**B**) of a mechanically sheared rectangular sensor that has been annealed. The images feature the 5 mm edge of a 12.7 × 5 mm sensor.

**Figure 3 sensors-22-00827-f003:**
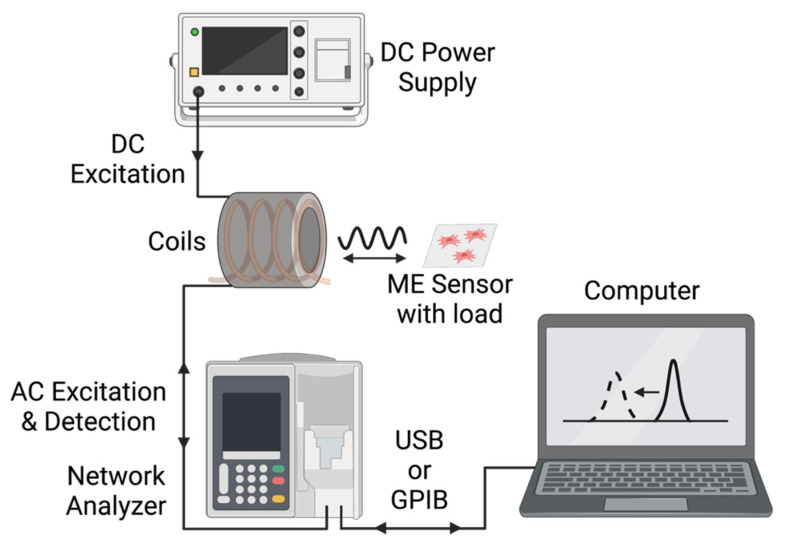
Block-diagram of the key instrumentation and interfaces in this experimental setup.

**Figure 4 sensors-22-00827-f004:**
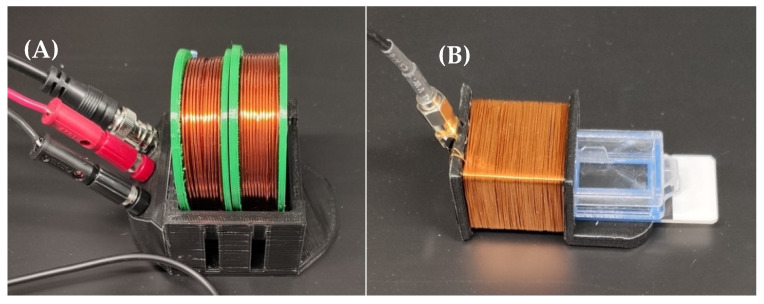
The DC bias coil (**A**) and detection coil (**B**) used in this experiment. During measurement, the detection coil is placed in the DC bias coil, such that the two coils are concentric. The exposed portion of the chamber slide is not used, as the sensor is only placed in the chamber that is completely inserted into the detection coil.

**Figure 5 sensors-22-00827-f005:**
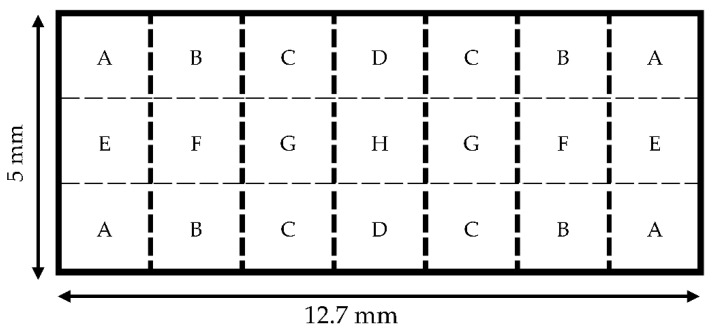
A diagram showing the 8 locations (A–H) where drops were deposited on the surface of each 12.7 mm × 5 mm sensor. Each letter represents a different location. With ‘H’ being at the origin center of the sensor, the rest of the letters (A–G) are mirrored across the four quadrants of the sensor surface to show symmetry.

**Figure 6 sensors-22-00827-f006:**
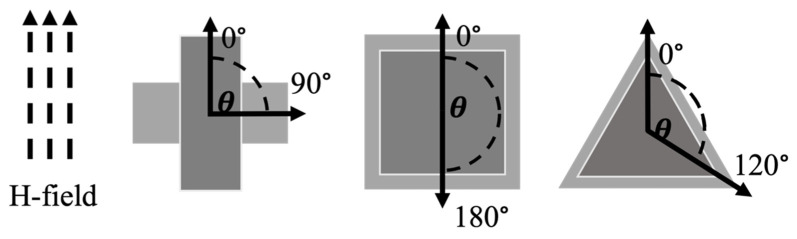
Schematic representation of magnetic field orientation and angles of maximum displacement in this study. The bias and activation fields are applied parallel and coincident with the 0° alignment of each shape, which is denoted by the darker color configuration. The lighter color configuration shows the same edge/corner that was initially aligned with 0° but is now aligned with the maximum displacement angle for that shape. The direction of applied fields remained the same throughout the experiments.

**Figure 7 sensors-22-00827-f007:**
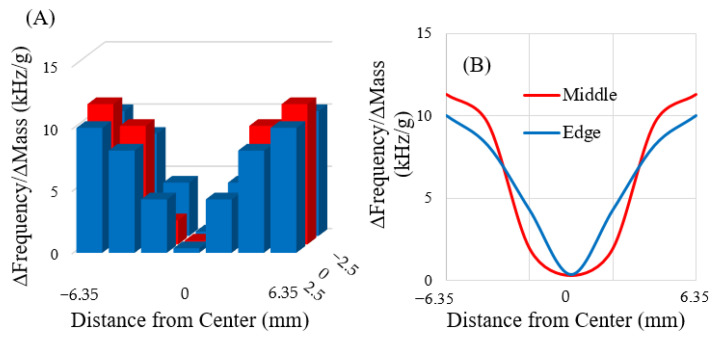
Charts illustrating the range of sensitivities that exist across the sensor surface in 3D (**A**) and as 2D lines along the length of the sensor (**B**). The rectangular sensors were assumed to behave symmetrically. The horizontal axes in both figures represent the entire length of the sensor, with the center of the sensor defined as the origin.

**Figure 8 sensors-22-00827-f008:**
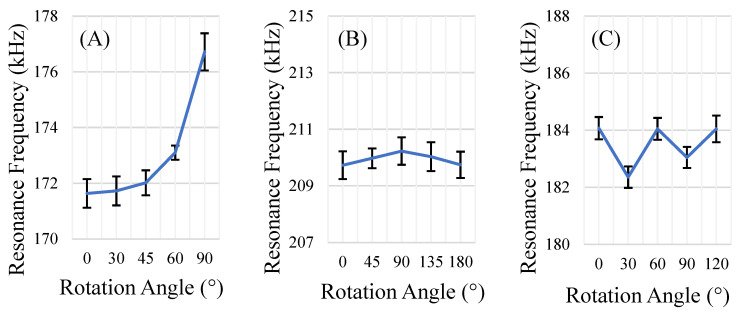
Changes in the sensor’s resonance frequency at varying directions of magnetic field. Three shapes of sensors were evaluated: rectangular (**A**, 12.7 mm × 5 mm), square (**B**, length = 12.7 mm), and equilateral triangle (**C**, base = 14.6 mm, height = 12.7 mm). Angular increments were selected based on the rotational symmetry of the shape. (*n* = 5; error = ± standard deviation).

**Figure 9 sensors-22-00827-f009:**
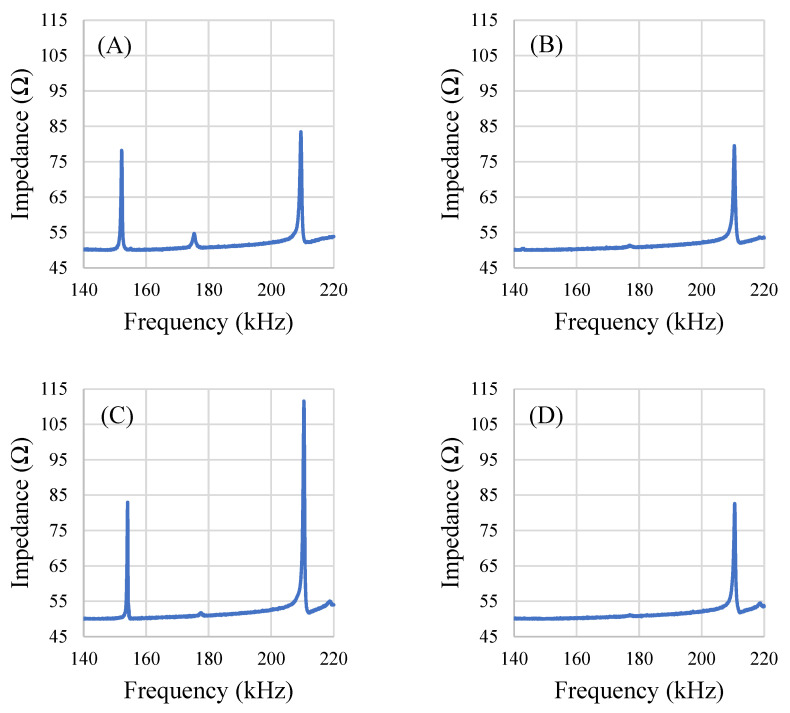
Plots of a single square sensor’s resonances at 0° (**A**), 45° (**B**), 90° (**C**), and 135° (**D**) rotations from the normal orientation.

**Figure 10 sensors-22-00827-f010:**
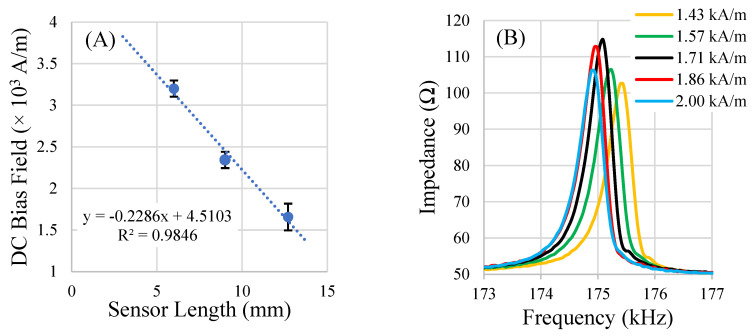
A plot of the results of the DC bias field optimization experiment (**A**) and an example of the under- and over-biasing effects on the resonance spectrum (optimal bias at 1.71 kA/m) (**B**). The sensors were all fabricated at an aspect ratio of 2.5 length over width. (*n* = 5; error = ± standard deviation).

**Figure 11 sensors-22-00827-f011:**
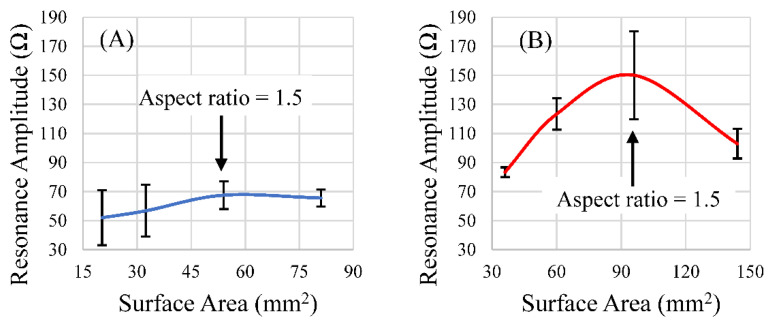
The responses of rectangular sensors fabricated at lengths 9 mm (**A**) and 12.7 mm (**B**) with varying widths were plotted against the surface area of those sensors. (*n* = 4; error = ± standard deviation).

**Figure 12 sensors-22-00827-f012:**
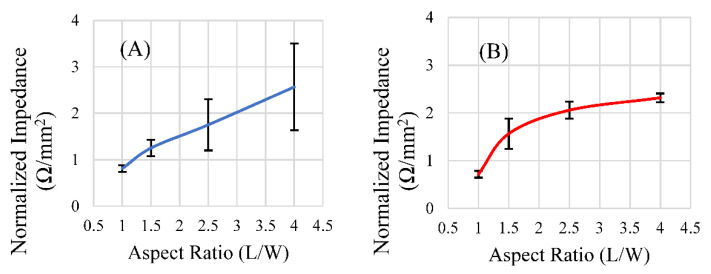
The response of rectangular sensors fabricated at lengths 9 mm (**A**) and 12.7 mm (**B**), normalized for their surface area, plotted against the aspect ratio of those sensors (*n* = 4, error = ± standard deviation).

## Data Availability

Not applicable.

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
