# Peer review of "Magnetoelastic Sensor Optimization for Improving Mass Monitoring"

_sensors, 2022, doi:10.3390/s22030827_

Round 1

Reviewer 1 Report

The manuscript examines how the geometry of the magnetoelastic sensor (i.e., size, shape, aspect ratio) impact the sensor’s resonance modes.  The goal is to optimize the geometry and measurement conditions for improving the sensor performance in detecting biological or chemical agents in dynamic and volumetric environments. In general, the research reported in the manuscript is technically sound with sufficient experimental evidence to support the main conclusions.  The fabrication of the sensors and system used to interrogation system (i.e., to read the sensor response) are described in sufficient detail that other research should be able to replicate the result.

There are, however, a number of critical issues that must be addressed before the manuscript is acceptable for publication.  First, the Abstract is very general.  It should clearly define the problem being investigated, state the research contribution, and summarize the key outcomes of any analysis or experimentation. Second, it is important that the authors relate their research to the broader literature and more recent studies on magnetoelastic sensors.  The present paper includes 29 references where 24 are from the senior author or a related group from Penn State (i.e., C.A. Grimes work).  As well, many of the papers listed are more than a decade old.  Third, it is not clear from Section 2.5 what the authors’ mean by optimization.  In general, the term optimization implies maximizing or minimizing a function. Fourth, many of the figures require additional explanation & proper labeling (e.g., Figs. 1, 4) or improved presentation/formatting (e.g., Figs. 5-10) for an archived journal paper.  In many of these figures the axes are not appropriately high-lighted, axis labels are too large and poorly placed on the graph, the lines are too thick for size of graph, etc.  Fifth, many of the graphs include error bars but the sample size and data statistics are not discussed.

Author Response

Please see the attached file for our point-by-point response to the reviewer's comments.

Reviewer 2 Report

Submitted manuscript is devoted to interesting subject – biomagnetic detection of the biological or chemical substances. This area is under rapid development and Sensors has a number of related Special Issues and MDPI is actively developing a corresponding subject (). In contract with this trend authors make quite short and limited introduction related to their own contributions or contributions to their co-authors. Actually, work presents a very clear alarming level of inappropriate self-citations and absence of mentions of contributions of the other groups. For example, Susniega et al. recently reported the study on  Real Time Monitoring of Calcium Oxalate Precipitation Reaction by Using Corrosion Resistant Magnetoelastic Resonance Sensors (Sensors 2020, 20(10), 2802; https://doi.org/10.3390/s20102802) and Sagasti et al. worked on Sagasti, A.; Gutiérrez, J.; Lasheras, A.; Barandiarán, J.M. Size dependence of the magnetoelastic properties of metallic glasses for actuation applications (Sensors 2019, 19, 4296  https://doi.org/10.3390/s19194296), Garcia-Arribas provided comparison of ME sensors with other types of magnetic sensors with capacity for biodetection in Sensor Applications of Soft Magnetic Materials Based on Magneto-Impedance, Magneto-Elastic Resonance and Magneto-Electricity (Sensors 2014, 14(5), 7602-7624; https://doi.org/10.3390/s140507602). Beato-Lopez discussed Giant Stress Impedance Magnetoelastic Sensors Employing Soft Magnetic Amorphous Ribbons (Materials 2020, 13(9), 2175; https://doi.org/10.3390/ma13092175). In any case, Introduction is rather short for 13 pages article and it gives quite limited approach to the physical basis of the improvement of the sensitivity of ME sensors oriented on the biodetection. The concept of the detectors with markers or without them is not even mentioned in the text.

Description of the ME detection was previously reported many times and instead authors could provide the general view of their sensitive elements or measuring device – manuscript does not show ant real sample with evidence of the quality of the surface, etc. Magnetic characterization is also absent.

Discussion is very superficial and needs to be deepen toward fundamental problems of this type of the detection. For instance, for following part - “In general, an aspect ratio near 1.5 was optimal for the rectangular sensor shapes of both lengths (Figures 9 and 10). After the data was normalized for differences in surface area between the two sets, it was interesting to note that the 9 mm sensors were capable of over twice the field output of the 12.7 mm sensors. We believe this difference was primarily due to the self-resonance of the detection coil and the difference in resonant frequencies of these sensors.” The argument can not be based on “believe”, self-resonance contribution can be evaluated and different coil used.

To what extent the quality of the edges (could the SEM or optical images be provided) can contribute to the results for sensitive elements with different geometries.

Round 2

Reviewer 2 Report

Submitted manuscript was significantly improved and Authors responded satisfactorily to some of my questions. However there are points whcich were not addressed.

  1. The SELF-citation level (including close collaborators) is still huge, it should not be above 20 or at most 25% for Q1 journal. Poor citation limits the interest to the journal and this particular work. It is especially alarming in a view of existing extended literature of biomagnetic sensing.
  2. I proposed to add general comment on the existence of different kinds of biomagnetic sensing. For example, label-free sensing do not limited by the ME sensors, there were attempts to develop different systems with ribbons for label free detection (magnetic impedance for instance). Appropriate overview and comparison ME sensors with the other types would be an advantage, including such points as price of the whole device and aspects of green production.
  3. Particular piece of commercial materials not always has the same properties as expected. Did you check the main parameters?  

Round 3

Reviewer 2 Report

Work was reasonably improved and it can be published in the present state.